# Muscle Recruitment and Asymmetry in Bilateral Shoulder Injury Prevention Exercises: A Cross-Sectional Comparison Between Tennis Players and Non-Tennis Players

**DOI:** 10.3390/healthcare13101153

**Published:** 2025-05-15

**Authors:** Maite Terré, Mònica Solana-Tramunt

**Affiliations:** Facultat de Psicologia, Ciències de l’Educació i l’Esport (FPCEE) Blanquerna, University Ramon Llull, 08022 Barcelona, Spain; maitetp@blanquerna.url.edu

**Keywords:** shoulder prevention exercises, muscle recruitment, scapular exercises, scapular retraction, mid trapezius, lower trapezius, overhead athletes

## Abstract

**Background/Objectives:** Shoulder injuries are common in overhead sports like tennis due to repetitive unilateral movements that can lead to muscle imbalances. This study aimed to compare muscle recruitment and asymmetry during bilateral shoulder injury prevention exercises (performed with both arms simultaneously) in tennis players versus non-tennis athletes. **Methods:** Thirty-nine athletes (sixteen tennis players, twenty-three non-tennis athletes) performed two bilateral scapular retraction exercises at 45° and 90° shoulder abduction. Surface electromyography (sEMG) recorded the activation of the middle and lower trapezius. Root Mean Square (RMS), peak RMS and muscle symmetry indices were analyzed. **Results:** Tennis players showed significantly lower trapezius activation, especially during prone retraction at 90°. Muscle symmetry was slightly higher in tennis players at 90°, but asymmetry increased at 45°, suggesting angle-specific adaptations. **Conclusions:** Repetitive asymmetric loading in tennis may reduce the activation of scapular stabilizers and contribute to muscular imbalances. Including targeted bilateral exercises in training may help improve scapular muscle function and reduce injury risk in overhead athletes.

## 1. Introduction

Sport disciplines are commonly categorized as either symmetrical or asymmetrical. Symmetrical sports, such as gymnastics and long-distance running, promote balanced development through bilateral movements, whereas asymmetrical sports, like tennis, fencing and javelin throwing, rely heavily on repetitive, unilateral actions. Despite the frequent use of this classification, it remains inconsistently defined and lacks a universally accepted framework [1].

In asymmetrical sports, athletes often develop muscular and structural imbalances, particularly favoring the dominant side [2,3]. While such adaptations may enhance sport-specific performance, they can also increase the risk of musculoskeletal injuries and negatively impact joint function. Conversely, symmetrical sports typically promote more uniform muscle recruitment across both limbs [4]. Nonetheless, individual biomechanics, training methods and sport-specific techniques can influence the development of asymmetries even in symmetrical disciplines [5].

Tennis is a prime example of an asymmetrical sport, characterized by repetitive actions such as serves and groundstrokes, which place high demands on the dominant shoulder and upper limb [6]. These repetitive loads can result in reduced internal rotation and increased external rotation of the glenohumeral joint, known as glenohumeral internal rotation deficit (GIRD) [7]. Overuse injuries associated with scapular dyskinesia, rotator cuff weakness and GIRD are frequently reported in overhead athletes [8,9].

These asymmetries not only increase injury risk but may also impair stroke consistency, trunk stability and inter-limb coordination, thereby reducing overall performance [5,10,11].

The scapular stabilizing muscles, particularly the middle and lower trapezius, play a critical role in shoulder function during high-load overhead movements. Imbalances in their activation, commonly seen in tennis players, may disrupt scapular mechanics and contribute to shoulder pathologies [11,12,13,14,15].

Understanding these neuromuscular adaptations is essential for the development of effective injury prevention strategies. Surface electromyography (sEMG) provides a non-invasive method to quantify muscle activation and asymmetries, and it has become a key tool in sport and rehabilitation science [5,13,16]. A muscle asymmetry occurs when the function between corresponding muscles on opposite sides of the body is greater than 20%. These inequalities can result from a variety of factors, including injury, unbalanced movement patterns, incorrect posture, or unequal load distribution during physical activities. The muscle symmetry index is used to quantify how similar two muscles are in terms of muscle activity: asymmetry, 0–79% similarity; limit, 80–89% similarity; and normal or symmetrical, 90–100% [16].

While the literature highlights GIRD and rotator cuff imbalances in overhead athletes, few studies have assessed upper limb asymmetries using preventive bilateral exercises. Moreover, most existing protocols are adapted from sports like baseball and do not fully address the unique demands of tennis. This is surprising given the high asymmetry and injury prevalence in tennis and the sport’s reliance on precise, high-force movements involving the upper limbs [13,17]. Therefore, this study aimed to investigate muscle activation patterns and asymmetries of the middle and lower trapezius during bilateral shoulder retraction exercises in athletes involved in asymmetric (tennis) versus symmetric sports. By analyzing electromyographic activity at 45° and 90° of shoulder abduction, we sought to determine whether repetitive unilateral loading in tennis leads to reduced activation and greater asymmetry. We hypothesized that tennis players would demonstrate lower trapezius activation and greater inter-limb asymmetry compared to non-tennis athletes during bilateral shoulder exercises.

## 2. Materials and Methods

### 2.1. Study Design

A repeated-measures cross-sectional between-group design was employed. The study was conducted in accordance with the STROBE reporting guidelines [18].

### 2.2. Study Population

The present study involved a total of 39 athletes from various sports disciplines. The athletes represented a wide range of sports disciplines. Specifically, 16 (41%) participants practiced tennis, making it the most common sport in the sample. The remaining athletes were distributed across various sports as follows: 8 (20.5%) were soccer players; 2 (5.1%) practiced athletics; 2 (5.1%) participated in futsal; and there were 10 (25.6%) participants each from triathlon, basketball, swimming, artistic gymnastics, judo, pole vaulting, dance, calisthenics, duathlon and gym training. Participants were classified into two groups: Non-tennis players (N–TPs) consisted of 23 athletes who do not engage in tennis, and tennis players (TPs) were composed of 16 athletes who actively participate in tennis. Both groups underwent the same physical assessments, and the analysis aimed to compare the differences between the two groups.

Demographic, anthropometric and descriptive measurements were collected following standardized anthropometric procedures to ensure accuracy and reliability (Table 1). This information was obtained through a self-reported questionnaire and verified during the assessment phase. Participants were also free of any pain or pathology that would prevent them from being able to perform shoulder external rotation exercises.

All participants fulfilled the inclusion criteria (Table 2) and were provided with both written and verbal explanations regarding the study procedures prior to the assessment session. Following the provision of detailed information, each participant signed an informed consent form, in accordance with last revision of the Declaration of Helsinki by the World Medical Association 2024 [19].

### 2.3. Outcomes

The main outcomes that have been considered to carry out an analysis are the Root Mean Square (RMS) (µV), Mean Maximum Amplitude (µV) of the 3 reps and muscle symmetries (%). The Muscle Symmetry Index used to quantify how similar two muscles are in terms of muscle activity was asymmetry, 0–79% similarity; limit, 80–89% similarity; and normal or symmetrical, 90–100% [10].

These variables were specifically analyzed in the lower trapezius and middle trapezius muscles, both on the right and left sides, considering the dominance of each player for a more accurate assessment of muscle asymmetry and its relationship with performance.

Electromyographic (EMG) data were recorded using the mDurance^®^ device (mDurance^®^ Solutions SL, Granada, Spain), which integrates an EMG Shimmer3 unit (Realtime Technologies Ltd., Dublin, Ireland). The mDurance^®^ system employs a validated surface EMG methodology designed for the accurate assessment of muscle activation. This advanced system combines user-friendly software with lightweight hardware, enhancing its practicality and accessibility for clinical and sports performance applications. The Shimmer3 unit within the system is a bipolar surface EMG sensor specifically engineered for the precise acquisition of muscle activity. Each unit is equipped with two recording channels, a sampling frequency of 1024 Hz, a bandwidth of 8.4 Hz and a 24-bit resolution, with a signal amplification range of 100 to 10,000 V/V. This configuration ensures high-quality data capture suitable for detailed analysis. The reliability and validity of the mDurance^®^ system for recording muscle activity during functional tasks have been established in prior research, demonstrating excellent interclass correlation coefficients (ICC = 0.916; 95% CI = 0.831–0.958) [20]. Such characteristics make this system an invaluable tool for researchers and practitioners aiming to assess neuromuscular performance in real-world scenarios. The mDurance Android app received the data from the Shimmer3 and sent it to a cloud service, where the data were stored, filtered and analyzed [21].

### 2.4. Procedure

After collecting descriptive information of the participants by a structured formula and dividing the groups into TPs and N–TPs, we proceed with the placement of electrodes.

#### Electrode Placement

All the participants were prepared for the EMG by shaving, scrubbing and cleaning the skin with alcohol to reduce impedance (<10 kΏ) [22]. The surface EMG electrodes were placed over the muscle’s bellies and in line with the muscle-fibers’ orientation of the lower trapezius right (LTR), lower trapezius left (LTL), middle trapezius right (MTR) and middle trapezius left (MTL). Self-adhesive 5 cm Dormo surface electrodes were placed on the muscle belly according to the SENIAM project recommendations [23] and with an interelectrode distance of 20 mm. The procedures for electrode placement and collection are shown in Table 3 and Figure 1, respectively.

### 2.5. Experimental Procedure

The participants performed two exercises proposed in the study to assess the activation pattern of the interscapular musculature during their execution and progression. Participants were instructed to refrain from high-intensity exercise for 24 h prior to both test sessions.

#### 2.5.1. Recording Muscle Activity

Each session lasted approximately 10 min per participant for each exercise, including the time required for electrode placement. The tests did not involve any particular risk. None of the procedures incurred any cost to the participants, and surface electromyography was used, which was non-invasive and comfortable. The setup allowed participants to perform all movements freely without restriction.

After electrode placement, detailed instructions were provided on the correct execution of the exercises. Participants performed a series of three contractions for each exercise, followed by a one-minute rest period. Each contraction was recorded for approximately 3–5 s, and data collection began after a 2 s stabilization phase to avoid noise at the start of the recording. Participants were instructed to maintain a regular breathing pattern, actively facilitating scapular retraction during inhalation and allowing relaxation during exhalation.

#### 2.5.2. Exercises

The first exercise focused on scapular retraction, breathing technique and core engagement by lifting the feet. During the concentric phase, the participants were instructed to pull their shoulder blades together, emphasizing scapular retraction while simultaneously lifting their feet off the ground. As the arms extended outward, they inhaled deeply. During the eccentric phase, the arms were lowered slowly and with control, while exhaling, as they returned to the starting position. The feet remained elevated throughout the movement, promoting core engagement and enhancing postural stability (Figure 2a). The exercises were initially performed at a 90° angle, followed by execution at 45°. This exercise was not evaluated using resistance bands.

The second exercise (Figure 2b) focused on scapular retraction and proper breathing technique while using a multi-loop resistance band (Flexvit Multi, Flexvit Bands, Haigerloch, Germany). The loops were adjusted to each participant’s wingspan to ensure the movement allowed for maximum scapular adduction. To determine the most appropriate loop tension for each individual, participants were instructed to perform maximal scapular retraction prior to initiating muscle activity recording. As in the previous exercise, the protocol began with assessments at 90°, followed by 45°.

Participants stood with their arms extended to the sides, holding the resistance band with both hands while maintaining proper shoulder alignment. The primary objective was to retract the shoulder blades by actively squeezing them together as the band was pulled outward. During the concentric phase (pulling the band), participants inhaled deeply while engaging the core for postural stability. In the eccentric phase (returning to the starting position), they exhaled in a controlled manner, maintaining scapular engagement throughout. The focus remained on achieving full scapular retraction while coordinating the breath with the movement (Figure 2b). This exercise was performed with only the resistance band.

### 2.6. Statistical Analysis

Descriptive statistics were computed for all variables and presented as means ± standard error (SE), where SE was defined as the standard deviation divided by the square root of the sample size. In addition, standard deviations were used to describe variability in quantitative variables, and frequencies were calculated for qualitative demographic variables. The normality of distributions was assessed using the Shapiro–Wilk or Kolmogorov–Smirnov tests, depending on sample size, with *p*-values greater than 0.05 considered indicative of normality. The homogeneity of variances was examined using Levene’s test. Group comparisons between tennis players (TPs) and non-tennis players (N–TPs) were conducted using independent-sample *t*-tests for all quantitative variables, including Root Mean Square (RMS), peak RMS and muscle symmetry indices. Analyses were performed at two levels of glenohumeral abduction: 45° and 90°. Comparative analyses were also conducted between two exercise conditions: (1) scapular retraction in a prone position and (2) scapular retraction in a standing position with elastic band resistance. For variables recorded for both exercises (e.g., 1_RMS_MT_D_90 vs. 2_RMS_MT_D_90), paired comparisons were performed across all participants and separately within each group (TPs and N-TPs) using independent-samples *t*-tests when assumptions were met. Effect sizes (ES) were calculated using Cohen’s d and interpreted as small (d < 0.3), moderate (0.3 ≤ d < 0.8), or large (d ≥ 0.8). Statistically significant differences were considered at *p* < 0.05. Comparisons reaching statistical significance or presenting meaningful effect sizes were marked with an asterisk (*) in all graphical representations. All statistical analyses were performed using JAMOVI software (version 2.5.4) and Python (v3.10).

## 3. Results

### 3.1. Exercise 1: Muscle Recruitment at 90° and 45°

A significant difference and large ES in muscle activation at 90° was observed between tennis players (TPs) and non-tennis players (N–TPs) for both dominant and non–dominant sides and for both MT and LT. The Student’s *T* Test revealed that N–TPs exhibited higher activation in MT and LT in both the dominant (D) and non-dominant side (ND) at 90° (415 µV vs. 255 µV for D, 438 µV vs. 253 µV for ND on MT; and 449 µV vs. 245 µV for D and 450 µV vs. 237 µV for ND on LT); see Figure 3.

No significant differences were observed in the activation of any of the trapezius muscles in D and ND at 45° (*p* > 0.05). However, the mean activation was higher in N–TPs (Figure 4).

### 3.2. Exercise 1: Muscle Symmetries at 90° and 45°

Both groups demonstrated symmetry indices that were below the normal threshold of 90%, with some values in the borderline or asymmetrical ranges (e.g., 75–82%). Very large effect sizes likely indicate internal asymmetry, particularly in the TP group. However, since *p*-values were not significant, these internal asymmetries are not statistically different between groups (Table 4).

### 3.3. Exercise 1: RMS Peaks at 90° and 45°

There were significant differences in RMS peaks at 90°. N–TPs exhibited higher RMS peaks: 960 µV vs. 568 µV for D and 974 µV vs. 564 µV for ND in MT; 944 µV vs. 529 µV. for D and 938 µV vs. 476 µV for ND in LT (Figure 5).

No significant differences were found in the highest RMS peaks during Exercise 1 developed at 45° (Figure 6).

### 3.4. Exercise 2: RMS at 90° and 45°

No significant differences were observed in the mean RMS at 90° across any of the muscles analyzed (*p* > 0.05). However, the mean activation was slightly higher in non-tennis players (Figure 7).

No significant differences were found in the mean RMS for any of the muscles analyzed at 45° (Figure 8).

### 3.5. Exercise 2: Muscle Symmetries at 90° and 45°

Both groups demonstrated symmetry indices that were below the normal threshold of 90%, with some values in the borderline or asymmetrical ranges (e.g., 75–82%). Very large effect sizes likely indicate internal asymmetry, particularly in the TP group. However, since *p*-values were not significant, these internal asymmetries are not statistically different between groups (Table 5).

### 3.6. Exercise 2: RMS Peaks at 90° and 45°

No significant differences were found in the peak RMS at 90° during the secod exercise (Figure 9).

No significant differences were found in the peak RMS at 45° during the second exercise (Figure 10).

### 3.7. Comparison Between Exercises and Muscle Groups

Significant differences in muscle activation were observed between the prone scapular retraction exercise (1) and the standing elastic-band retraction exercise (2) in both tennis players (TPs) and non-tennis players (N-TPs), particularly for the middle trapezius (MT) and lower trapezius (LT) at both 45° and 90° of glenohumeral abduction.

In the TP group, Exercise 1 elicited significantly greater activation of the MT and LT at 90° compared to Exercise 2 (*p* < 0.001), with large effect sizes (Cohen’s d = 1.49 and 1.73, respectively). At 45°, LT activation was also significantly higher during the prone exercise (*p* = 0.007, d = 1.04), whereas no significant difference was found for MT activation (*p* = 0.145). In the N–TP group, a significant difference was observed only for the MT at 45°, with higher activation during the standing exercise (*p* = 0.048, d = 0.73) (Figure 11).

## 4. Discussion

This study aimed to examine shoulder muscle recruitment and asymmetry differences between tennis players (TPs) and non-tennis players (N–TPs) during bilateral scapular retraction exercises at 90° and 45° of abduction. The hypothesis—stating that TPs would demonstrate lower trapezius activation and greater inter-limb asymmetry—was partially supported by the results.

### 4.1. Muscle Recruitment in Exercise 1: Prone Bilateral Scapular Retraction

During prone bilateral scapular retraction (exercise 1) at 90°, N–TPs exhibited significantly higher RMS and peak RMS values in both the middle and lower trapezius muscles across dominant and non-dominant sides. This suggests that N–TPs may recruit scapular stabilizers more effectively during isolated prone retraction tasks. In contrast, TPs likely experience sport-specific neuromuscular adaptations due to repetitive unilateral movements, resulting in the reduced activation of these stabilizing muscles [13,24,25].

At 45°, no statistically significant differences were observed, although N–TPs showed slightly higher average activation. This may reflect the biomechanical similarity of this angle to tennis strokes, in which the activation of surrounding muscles may be shared or compensated. Additionally, the prone position reduces postural demands, allowing for the more standardized recruitment of scapular muscles across groups [2,3,26].

Regarding symmetry, no significant group differences were found. However, both groups exhibited values below 90%, suggesting mild-to-moderate asymmetry. TPs demonstrated slightly greater symmetry at 90°, possibly due to adaptations in the dominant limb. In contrast, N–TPs showed greater symmetry at 45°, highlighting potential angle-specific neuromuscular imbalances in TPs related to sport-specific patterns [27,28]. Along the same lines, it could be suggested that TPs exhibit signs of chronic fatigue or employ compensatory mechanisms that may reduce the activation of the lower and middle trapezius muscles. Unfortunately, to the best of our knowledge, there are no previous studies available for direct comparison regarding the activation patterns of the lower or middle trapezius. Existing research has primarily focused on the activity of the supraspinatus, infraspinatus, posterior deltoid and pectoralis major during specific sport-related movements rather than during preventive exercises such as those used in our study [9]. The prone position offers a more stable and balanced base of support, allowing for greater isolation and targeted recruitment of specific scapular stabilizers. In contrast, during the standing exercise, the body must actively maintain balance and posture, which demands greater involvement of trunk and pelvic stabilizing muscles. This additional requirement may redirect neuromuscular engagement toward supporting muscles—such as the deltoids or trunk stabilizers—thereby diminishing the relative activation of the middle and lower trapezius muscles compared to the prone condition [28]. Gravity also influences the mechanics of the exercise, as standing requires the muscles to counteract a different distribution of forces, necessitating greater body stabilization during movement. This shift in mechanical demand may affect the specific activation of the scapular musculature [28,29].

### 4.2. Muscle Recruitment in Exercise 2: Standing Bilateral Scapular Retraction with an Elastic Band

No significant differences in mean or peak RMS were detected at either 90° or 45°, although N–TPs again tended to show slightly higher values. The standing posture introduces additional balance and core demands, which may alter neuromuscular strategies. TPs may rely on compensatory patterns involving prime movers like the deltoid, reducing reliance on the trapezius during less dynamic, bilateral tasks [29]. Furthermore, in this position, TPs may compensate for the movement with the deltoid, which could result in a reduced activation of the trapezius. The deltoid plays a role in controlling scapular retraction and arm abduction. However, instead of maintaining primary activation in the scapular stabilizers, such as the middle and lower trapezius, TPs may rely on the deltoid [30,31]. This is particularly relevant for TPs, who perform repetitive, high-load, unilateral movements, which can lead to the suboptimal activation of the scapular stabilizers, such as the middle and lower trapezius. This imbalance in muscle activation and neuromuscular control may increase the risk of muscle imbalances and long-term shoulder injuries [2,32]. Muscle symmetry values during this exercise also showed no significant differences between groups, though TPs displayed more asymmetry at 45°, an angle highly relevant to tennis strokes. This may reflect a muscular imbalance developed over years of unilateral loading. Moreover, asymmetries between 10% and 15% are often associated with a higher risk of injury and reduced performance [10,33].

Tennis players, especially at 45°, might show more pronounced asymmetries because the muscles involved in tennis (such as the deltoids, rotator cuff and scapular stabilizers) may develop differently on each side of the body. This asymmetry is due to the sport’s high reliance on one side of the body for specific actions, such as serving or forehand/backhand strokes. Over time, the body adapts, and these adaptations may cause an imbalance in muscle symmetry. At 90°, the differences may be less pronounced because the movement is less specific to the sport, thus minimizing the effect of this imbalance.

A crucial factor in altered scapular movement and/or shoulder injuries is the imbalance in scapular muscle activation, which includes the reduced activation of the lower trapezius or middle trapezius. Functionally, the middle trapezius and lower trapezius contribute to scapular stability by limiting unnecessary vertical and horizontal displacements of the scapula and maintaining its proper positioning [15,34].

### 4.3. Between-Exercise Comparisons

TPs demonstrated significantly greater MT and LT activation during the prone exercise (Exercise 1) compared to standing retraction (Exercise 2), especially at 90°, with large effect sizes (d > 1.0). This may reflect the improved isolation of scapular stabilizers in prone conditions [28,29,33]. In contrast, N–TPs showed more variable responses, with slightly higher MT activation during standing retraction at 45° (*p* = 0.048, d = 0.73), likely due to their more generalised neuromuscular training backgrounds. These findings suggest that posture and shoulder abduction angle strongly influence scapular recruitment patterns, particularly in athletes with different sport-specific adaptations [35]. In the second exercise (standing scapular retraction), tennis players, who often perform dynamic, high-velocity movements, may have a higher degree of muscle recruitment efficiency in sport-specific actions, but they might not activate certain stabilizing muscles in exercises like scapular retraction [28]. These muscles, including the middle and lower trapezius, might not be as actively engaged in TPs due to the specificity of their training, leading to a lower RMS in comparison to non-tennis players.

Overall, prone scapular retraction (Exercise 1) tended to elicit greater muscle activation in variables with significant or moderate effects, especially in the TP group. This may reflect their greater familiarity with scapular control in closed kinetic chain positions, potentially due to sport-specific adaptations. In contrast, the N-TP group showed less consistent differences between exercises, suggesting a more generalized neuromuscular response. These findings support the inclusion of both prone and elastic band retraction exercises in shoulder rehabilitation programs while tailoring angle and modality based on athletic background and neuromuscular demands. More specifically, the present findings suggest that the prone scapular retraction exercise (Exercise 1) is more effective at activating the middle and lower trapezius muscles in tennis players, particularly at 90° glenohumeral abduction. This pattern may be attributed to sport-specific adaptations associated with repeated overhead movements in tennis, which are known to alter scapular muscle recruitment patterns [34,36]. Prone shoulder horizontal abduction is commonly used to strengthen the lower trapezius. Previous studies comparing lower trapezius activation across various exercises have suggested that performing this exercise in alignment with the lower trapezius muscle fibers may enhance its activation [27].

The large effect sizes observed in the TP group (Cohen’s d > 1.0) reinforce the efficacy of prone-based retraction exercises in eliciting the targeted activation of key scapular stabilizers. The lower trapezius plays a crucial role in maintaining scapular posterior tilt and external rotation during shoulder elevation—a function critical to reducing subacromial impingement risk [36,37]. In contrast, the N-TP group demonstrated only a moderate effect in favor of the standing exercise at 45° for MT activation. This suggests that non-specialist athletes may benefit from more general approaches, while overhead athletes may require more specific positioning (e.g., prone, high abduction) to fully engage the scapular musculature.

The imbalance between upper trapezius overactivity and the under-recruitment of the lower trapezius has been linked to shoulder dysfunction and injury in overhead athletes [38]. Given the superior activation of LT and MT observed in prone exercises at 90°, incorporating such exercises into preventive programs may enhance scapular control, reduce the risk of impingement and improve kinetic chain integration during service and overhead strokes [39,40]. Therefore, we recommend the integration of prone scapular retraction exercises at 90° abduction into shoulder injury prevention protocols for tennis players, particularly during the pre-season and preparatory phases. Additionally, progression to functional standing variations may be appropriate in later stages of training or for maintenance in less experienced athletes.

### 4.4. Limitations

This study presents several limitations that should be considered when interpreting the results. First, the relatively small sample size (*n* = 39) and the imbalance between groups (16 TPs vs. 23 N–TPs) may limit the statistical power and generalizability of the findings. At the time of data collection, the sample size (*n* = 39) was determined based on the availability of eligible athletes from local clubs and academies who met the inclusion criteria for both the tennis and non-tennis groups. However, we acknowledge that a formal a priori power analysis was not performed prior to recruitment. To address this, a post hoc power analysis was conducted using G*Power (version 3.1.9.7), based on the observed effect sizes for RMS activation differences between groups (e.g., d = 1.49 for the lower trapezius at 90°). The achieved power (1–β) for this comparison exceeded 0.90, indicating a very high probability of detecting true differences with the current sample size.

Second, the heterogeneity within the non-tennis group—comprising athletes from diverse sports—introduces potential variability in neuromuscular adaptations that could influence EMG results. Third, muscle activation was assessed only in the middle and lower trapezius. Other key stabilizers (e.g., supraspinatus, infraspinatus and rhomboids) were not included due to technical limitations of surface EMG, which may reduce the completeness of the neuromuscular profile. Additionally, prior training load and fatigue status were not controlled or recorded, potentially affecting activation patterns. Lastly, participants were not stratified by age or playing level, both of which can significantly impact neuromuscular adaptation. These factors warrant further study with larger, more homogenous samples and refined methodological controls. Acute and chronic training loads have been shown to affect muscle recruitment strategies, potentially altering the activation of scapular stabilizers such as the trapezius, infraspinatus and rhomboids. Insufficient recovery or accumulated fatigue from prior training sessions or during the session could lead to compensatory activation patterns or a decrease in muscle performance during testing [41].

Finally, this study did not consider the age of the participants or their level of tennis proficiency, which could have influenced the results. The broad sample, without specific stratification by age and skill level, may limit the generalizability of the findings to specific populations, such as elite TPs or developing athletes. Experience has been interpreted as functional specialization, which leads to structural and functional adaptations in the neuromuscular system due to long-term tennis practice. Although the emergence of functional specialization initially suggests a potential advantage (e.g., more powerful and faster stroke actions), evidence indicates a decline in physical performance if asymmetry exceeds a certain threshold. Despite the documented negative effects of neuromuscular asymmetry on lower limb athletic performance, there is a lack of studies investigating its impact on upper limb performance. This is particularly surprising, as neuromuscular asymmetry in upper-quarter mobility and stability has previously been associated with an increased risk (i.e., a risk ratio of 1.2) of sustaining a time-loss musculoskeletal injury in the future [33]. To enhance the validity and relevance of future research, it would be beneficial to define a more specific age range and classify participants according to their playing level. This approach would allow for a more precise analysis of individual differences and their impact on shoulder mechanics.

### 4.5. Practical Applications

From a practical standpoint, the findings highlight the importance of tailoring injury prevention and strength programs to sport-specific neuromuscular profiles. For tennis players, prone scapular retraction at 90° abduction appears most effective in activating the lower and middle trapezius muscles, which is key for maintaining scapular stability and preventing shoulder pathologies. EMG analysis can be strategically used to select exercises that elicit high activation in the target muscles [22]. Coaches should prioritize prone-based training in pre-season and preparatory phases and later progress to standing variations for functional carryover. Screening at 45° and 90° can help detect angle-specific deficits, guiding corrective strategies. Optimizing scapular muscle function could reduce overuse injuries and enhance stroke efficiency, particularly by restoring balance in musculature prone to asymmetrical development due to tennis-specific patterns. This could enable the development of more precise and effective training regimens, specifically tailored to enhance the relevant muscle groups for tennis players. By using EMG to assess muscle activation during various exercises, we can optimize training programs that focus on the biomechanics of the sport, ensuring that exercises are aligned with the specific demands of tennis. This approach not only ensures more efficient muscle activation but also helps in reducing the risk of injury by promoting balanced muscle development, which is critical for improving athletic performance. Furthermore, tailoring training to the biomechanical demands of tennis allows for more sport-specific conditioning, which is essential for enhancing both strength and movement efficiency on the court. Moreover, coaches could choose between assessing at 45° or 90°, depending on the technical deficits observed in their athletes, thus helping professional athletes avoid overuse problems and enabling non-professional TPs to guarantee their correct technical development.

## 5. Conclusions

This study highlights the importance of targeted trapezius activation in injury prevention programs for tennis players. Compared to athletes in symmetrical sports, tennis players demonstrated lower activation of the middle and lower trapezius during bilateral scapular exercises—likely a result of repetitive unilateral loading.

These findings support incorporating specific exercises to improve neuromuscular balance and address asymmetries arising from tennis-specific demands. Muscle symmetry analysis during these exercises provides valuable insights for tailoring preventive training. Coaches and practitioners should consider shoulder abduction angle and exercise modality when designing protocols, especially for athletes exposed to chronic asymmetrical loads.

## Figures and Tables

**Figure 1 healthcare-13-01153-f001:**
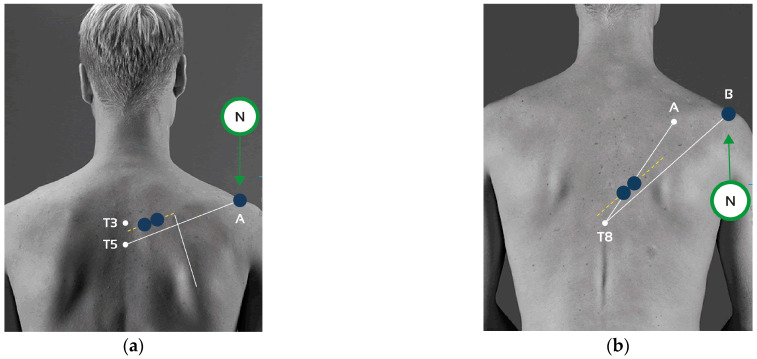
Placement of electrodes: (**a**) Middle trapezius (MT); T3: Spinous process of the 3rd thoracic vertebra; T5: Spinous process of the 5th thoracic vertebra; Line A–T5: Direction of the line between T5 and the acromion; N: Neutral electrode on the acromion surface. (**b**) Lower trapezius (LT); T8: Spinous process of the 8th thoracic vertebra; N: Neutral electrode on the acromion surface; Line A–T8: line from the trigonum spinae to the 8th thoracic vertebra; Line B–T8: line between T8 and the acromion.

**Figure 2 healthcare-13-01153-f002:**
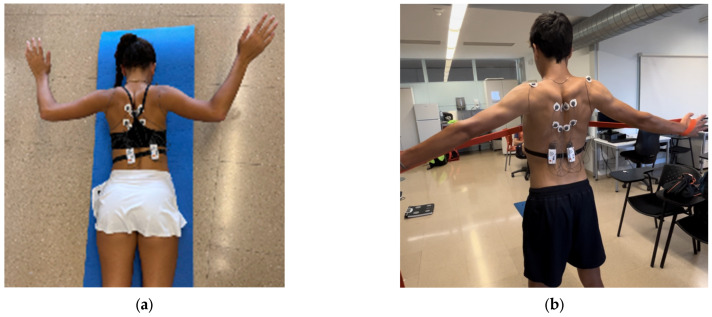
(**a**) Exercise 1: Prone lying bilateral adduction in 90°; (**b**) Exercise 2: Standing bilateral adduction in 90°.

**Figure 3 healthcare-13-01153-f003:**
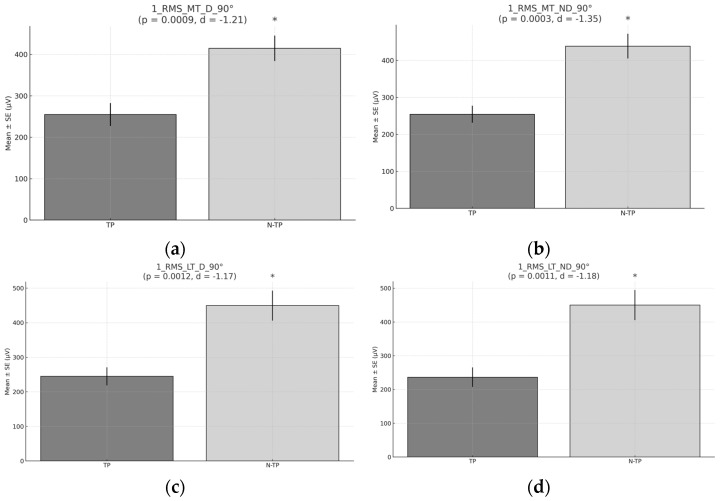
Mean RMS (µV) during Exercise 1 developed at 90°: (**a**) RMS for middle trapezius (MT) for the dominant side (D) comparison between tennis players (TPs) and non-tennis players (N–TPs). (**b**) RMS for middle trapezius (MT) for the non-dominant (ND) side comparison between TPs and N–TPs. (**c**) RMS for lthe ower trapezius (LT) for the dominant side (D) comparison between TPs and N–TPs (**d**) RMS for the lower trapezius (LT) for the non-dominant side (ND) comparison between TPs and N–TPs. Asterisks (*) denote statistically significant differences (*p* < 0.05). d = Cohen’s d ES.

**Figure 4 healthcare-13-01153-f004:**
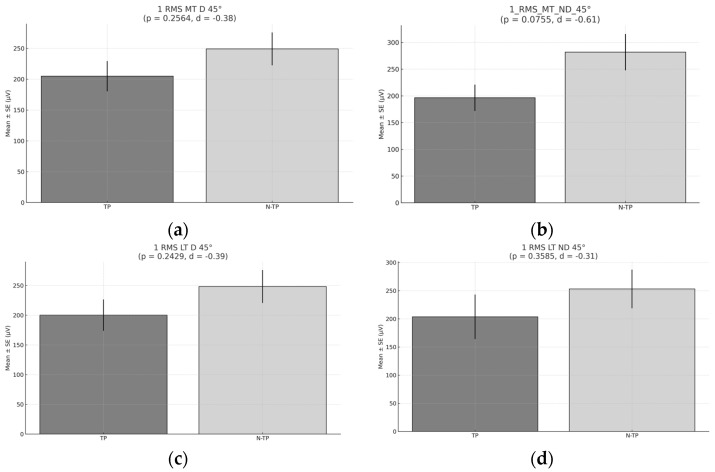
RMS during Exercise 1, 45°. (**a**) RMS for the middle trapezius (MT) for the dominant side (D) comparison between tennis players (TPs) and non-tennis players (N–TPs). (**b**) RMS for the middle trapezius (MT) for the non-dominant (ND) side comparison between tennis players and non-tennis players. (**c**) RMS for the lower trapezius (LT) for the dominant side (D) comparison between tennis players and non-tennis players. (**d**) RMS for the lower trapezius (LT) for the non-dominant side (ND) comparison between TPs and N–TPs.

**Figure 5 healthcare-13-01153-f005:**
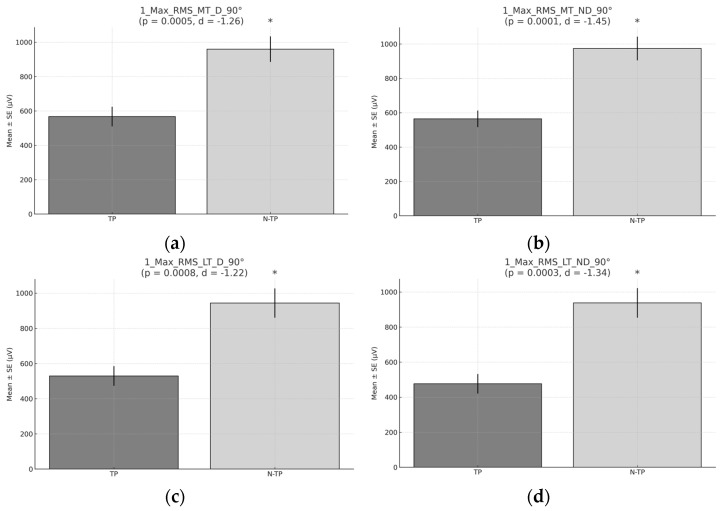
The peak RMS during Exercise 1 at 90°. (**a**) Peak RMS for the middle trapezius (MT) for the dominant side (D) comparison between tennis players and non-tennis players. (**b**) Peak RMS for the middle trapezius (MT) for non-dominant (ND) side comparison between tennis players and non-tennis players. (**c**) Peak RMS for the lower trapezius (LT) for the dominant side (D) comparison between tennis players and non-tennis players. (**d**) Peak RMS for the lower trapezius (LT) for the non-dominant side (ND) comparison between tennis players (TPs) and non-tennis players (N–TPs). Asterisks (*) denote statistically significant differences (*p* < 0.05).

**Figure 6 healthcare-13-01153-f006:**
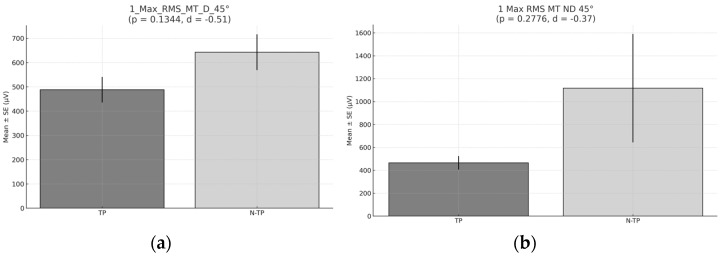
The peak RMS during Exercise 1 at 45°. (**a**) Peak RMS for the middle trapezius (MT) for the dominant side (D) comparison between tennis players and non-tennis players. (**b**) Peak RMS for the middle trapezius (MT) for the non-dominant (ND) side comparison between tennis players and non-tennis players. (**c**) Peak RMS for the lower trapezius (LT) for the dominant side (D) peak comparison between tennis players and non-tennis players. (**d**) RMS for the lower trapezius (LT) for the non-dominant side (ND) comparison between tennis players (TPs) and non-tennis players (N–TPs). Asterisks (*) denote statistically significant differences (*p* < 0.05).

**Figure 7 healthcare-13-01153-f007:**
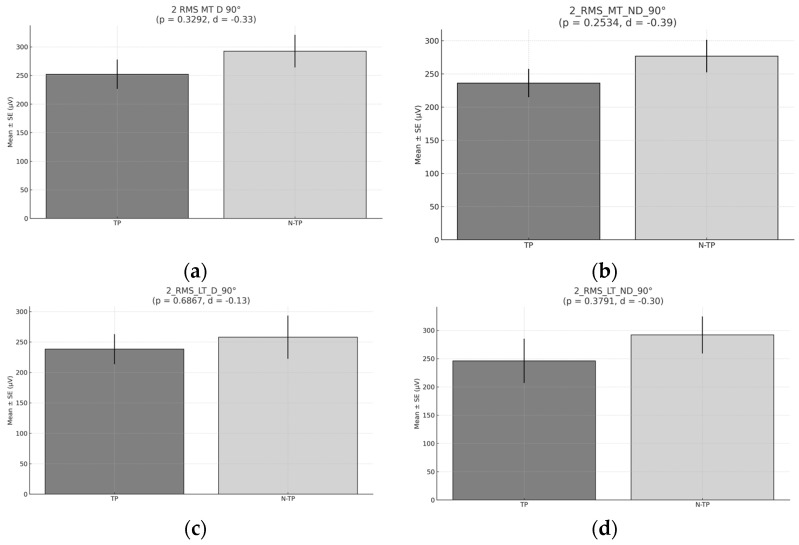
RMS during Exercise 2 at 90°. (**a**) RMS for the middle trapezius (MT) for the dominant side (D) comparison between tennis players and non-tennis players. (**b**) RMS for the middle trapezius (MT) for the non-dominant (ND) side comparison between tennis players and non-tennis players. (**c**) RMS for the lower trapezius (LT) for the dominant side (D) comparison between tennis players and non-tennis players. (**d**) RMS for the lower trapezius (LT) for the non-dominant side (ND) comparison between tennis players (TPs) and non-tennis players (N–TPs).

**Figure 8 healthcare-13-01153-f008:**
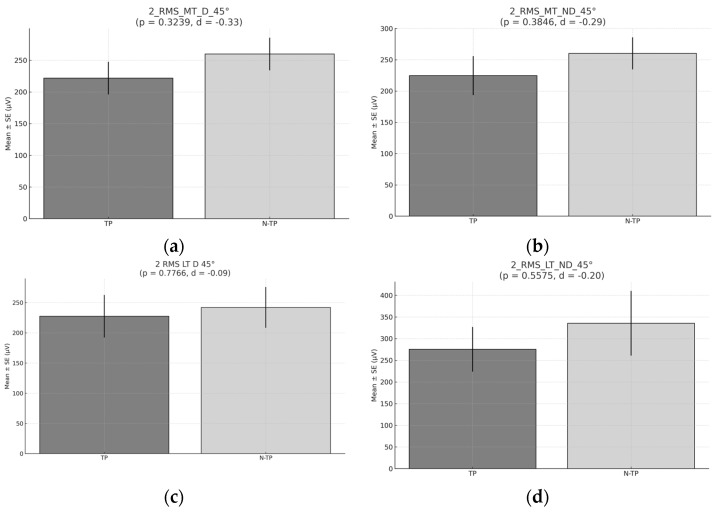
RMS during Exercise 1 45° (**a**) RMS for Middle trapezius (MT) for Dominant side (D) comparison between tennis players and non-tennis players. (**b**) RMS for Middle trapezius (MT) for Non–Dominant (ND) side comparison between tennis players and non-tennis players (**c**) RMS for lower trapezius (LT) for Dominant side (D) comparison between tennis players and non-tennis players (**d**) RMS for lower trapezius (LT) for Non–Dominant side (ND) comparison between tennis players (TP) and non-tennis players (N–TP).

**Figure 9 healthcare-13-01153-f009:**
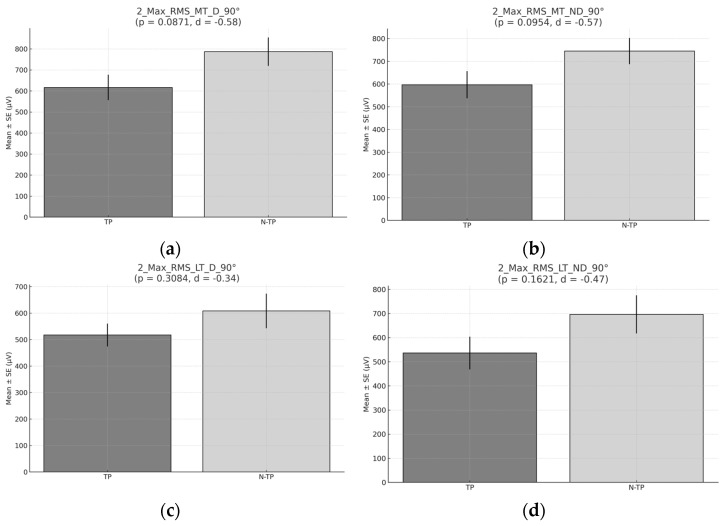
The peak RMS during Exercise 2 at 90°. (**a**) Peak RMS for the middle trapezius (MT) for the dominant side (D) comparison between tennis players and non-tennis players. (**b**) Peak RMS for the middle trapezius (MT) for the non-dominant (ND) side comparison between tennis players and non-tennis players. (**c**) Peak RMS for the lower trapezius (LT) for the dominant side (D) comparison between tennis players and non-tennis players. (**d**) Peak RMS for the lower trapezius (LT) for the non-dominant side (ND) comparison between tennis players (TPs) and non-tennis players (N–TPs).

**Figure 10 healthcare-13-01153-f010:**
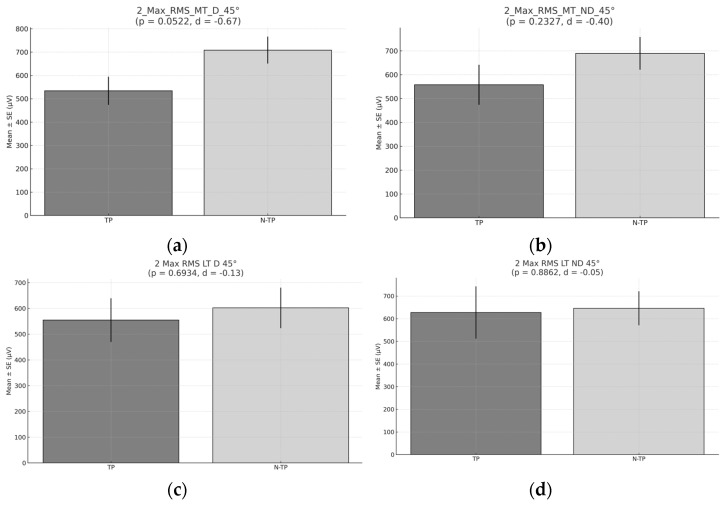
The peak RMS during Exercise 1 at 45°. (**a**) Peak RMS for the middle trapezius (MT) for the dominant side (D) comparison between tennis players and non-tennis players. (**b**) Peak RMS for the middle trapezius (MT) for the non-dominant (ND) side comparison between tennis players and non-tennis players. (**c**) Peak RMS for the lower trapezius (LT) for the dominant side (D) peak comparison between tennis players and non-tennis players. (**d**) RMS for the lower trapezius (LT) for the non-dominant side (ND) comparison between tennis players (TPs) and non-tennis players (N–TPs).

**Figure 11 healthcare-13-01153-f011:**
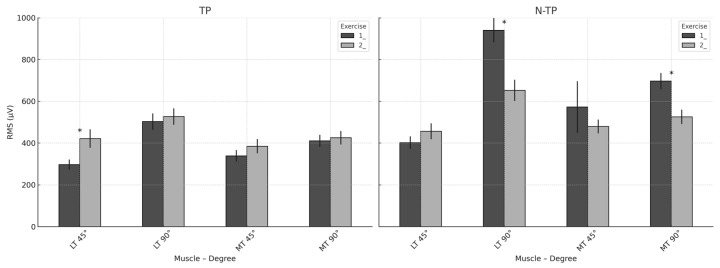
Comparison of mean RMS muscle activation (μV) between prone (1) and standing (2) scapular retraction exercises in tennis players (TPs) and non-tennis players (N-TPs). Each bar represents the average activation of the middle trapezius (MT) and lower trapezius (LT) muscles at 45° and 90° of glenohumeral abduction, with error bars indicating ±1 standard error of the mean (SEM). Asterisks (*) denote statistically significant differences between exercises 1 and 2 within each group (*p* < 0.05, independent-sample *t*-test).

**Table 1 healthcare-13-01153-t001:** Descriptive characteristics of the participants.

Descriptives	TP	N–TP
*n*	16	23
Age (year), mean (SD)	22.0 (7.7)	21.5 (8.0)
Gender, *n* female (%)	3 (18.7%)	3 (13.0%)
Body mass, mean (SD)	73.0 (9.3)	75.5 (9.1)
Height m, mean (SD)	1.8 (0.07)	1.8 (0.08)
BMI, mean (SD)	23.6 (2.2)	24.0 (2.0)
Shoulder injury history, *n* yes (%)	3 (18.7%)	7 (30.4%)
Shoulder pain during practice, *n* yes (%)	3 (13.0%)	7 (30.4%)
VAS (0–10) of pain, *n* (%)	0, 13 (75%)	0, 16 (69.5%)
	2, 1 (6.2%)	2, 3 (13.0%)
	3, 0 (0.0%)	3, 0 (0.0%)
	4, 1 (6.2%)	4, 1 (4.3%)
	5, 1 (6.2%)	5, 2 (8.6%)
Dominant right limb, *n* yes (%)	16 (100%)	21 (91.3%)
Sport experience (years), mean (SD)	12 (8.0)	10 (8.8)
Competition experience (years), mean (SD)	10 (4.8)	9.5 (5.5)

**Table 2 healthcare-13-01153-t002:** Inclusion criteria.

Inclusion Criteria
1. Tennis players must be part of a high-performance tennis club/academy. 2. Non-tennis players must never be involved in racquet sports. 3. Players must have a minimum of 2 years of experience in their respective sports. 4. Players must not suffer from any ailment or discomfort that would prevent him/her from performing the shoulder prevention exercises. 5. Players must not be taking medications throughout the study. 6. Players must have been free of musculoskeletal injuries over the previous three months.

**Table 3 healthcare-13-01153-t003:** Electrode placement.

Muscle	Electrode Placement	Orientation
Lower trapezius (LT)	At 2/3 on the line from the trigonum spinae to the 8th thoracic vertebra	The line between T8 and the acromion
Middle trapezius (MT)	At 50% between the medial border of the scapula and the spine, at the level of T3	Direction of the line between T5 and the acromion

Lower trapezius (LT); T8: spinous process of the 8th thoracic vertebra; Middle trapezius (MT); T3: spinous process of the 3rd thoracic vertebra; T5: spinous process of the 5th thoracic vertebra.

**Table 4 healthcare-13-01153-t004:** Muscle symmetries during exercise 1: bilateral prone adduction at 90° and 45°.

	Group	Mean (SD)	ES	*p*
MT_90°	Tennis player	79.9 (12.2)	3.16	0.620
	Non-tennis player	82.0 (12.8)	2.68	0.654
LT_90°	Tennis player	82.0 (13.0)	3.34	0.679
	Non-tennis player	80.0 (15.8)	3.29	0.917
MT_45°	Tennis player	80.4 (12.9)	3.34	0.277
	Non-tennis player	75.7 (13.0)	2.71	0.262
LT_45°	Tennis player	79.7 (15.0)	3.88	0.517
	Non-tennis player	76.1 (17.8)	3.72	0.622

Middle trapezius (MT); Lower trapezius (LT); Standard deviation (SD); ES = Effect size (Cohen’s d); *p* = statistical significance.

**Table 5 healthcare-13-01153-t005:** Muscle symmetries during Exercise 2: standing bilateral adduction at 90° and 45°.

	Group	Mean (SD)	ES	*p*
MT_90°	Tennis player	77.4 (15.6)	4.04	0.939
	Non-tennis player	77.7 (11.7)	2.43	0.8111
LT_90°	Tennis player	76.2 (12.7)	3.28	0.808
	Non-tennis player	77.5 (18.2)	3.80	0.437
MT_45°	Tennis player	76.9 (14.7)	3.78	0.882
	Non-tennis player	76.2 (14.4)	3.00	0.917
LT_45°	Tennis player	75.9 (18.5)	4.76	0.679
	Non-tennis player	78.3 (17.0)	3.55	0.834

Middle trapezius (MT); Lower trapezius (LT); Standard deviation (SD); ES = Effect size (Cohen’s d); *p* = statistical significance.

## Data Availability

The data presented in this study are available on request from the corresponding author. The data are not publicly available due to ethical restrictions related to participant confidentiality and the terms of informed consent.

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
