# Peer review of "Muscle Recruitment and Asymmetry in Bilateral Shoulder Injury Prevention Exercises: A Cross-Sectional Comparison Between Tennis Players and Non-Tennis Players"

_healthcare, 2025, doi:10.3390/healthcare13101153_

Round 1
Reviewer 1 Report
Comments and Suggestions for Authors
Title
The dot next to the title should be deleted.
Abstract
The conclusion in the abstract is too broad. You should only present conclusions and inferences based on your own research.
Introduction
The introduction is prepared in a very long and complicated way. First of all, the introduction should be rewritten in a simpler and more understandable way. During the rewriting, care should be taken to use more up-to-date sources. Using so many sources will also harm the originality of the research.
In addition, another important issue; the problem status of the research, the research question and the importance of the research should be rethought and expressed in a more understandable way.
Method
The material and methodology of the study are appropriate. However, there is a serious and important problem.
Is the number of participants included in the study sufficient and appropriate for this study? How was the number calculated? Why should a power analysis be performed? I strongly recommend a power analysis.
Also, the non-tennis group seems very scattered, why was such a wide range preferred? It should be explained.
Results
Abbreviations included in tables should be presented below the table. If ES represents effect size, statistical analyses may be incorrect. While p-value does not indicate a significant result, effect sizes are very large. All analyses should be reviewed.
The findings were not well presented.
Discussion
In the first part of the discussion, it was not stated whether the hypothesis was accepted or not.
The discussion section contains unnecessary details. It is written too long, too complicated. There is no integrity of the subject. Therefore, it is not understood. It should be rewritten and should include only current sources. It should also be short, simple and understandable for the purpose.
In addition, every research has its strengths, weaknesses and limitations. These are not included in this research. In particular, there is no section on limitations. This is an important issue.
In addition, no strong recommendations were made for some protic applications based on the research results.
Author Response
Comment#1: Title: The dot next to the title should be deleted.
Authors Response#1: We thank the reviewer for pointing this out. The dot next to the manuscript title has been removed to align with the journal’s formatting guidelines.
Comment#2: Abstract: The conclusion in the abstract is too broad. You should only present conclusions and inferences based on your own research.
Authors Response#2: We appreciate this observation and have revised the conclusion in the abstract to focus exclusively on the findings of our study. Overly general statements have been removed, and the conclusion now reflects only the specific outcomes derived from the current research.
Comment#3: Introduction: The introduction is prepared in a very long and complicated way. First of all, the introduction should be rewritten in a simpler and more understandable way. During the rewriting, care should be taken to use more up-to-date sources. Using so many sources will also harm the originality of the research. In addition, another important issue; the problem status of the research, the research question and the importance of the research should be rethought and expressed in a more understandable way.
Authors Response#3: We fully agree with the reviewer’s suggestions and have rewritten the Introduction section to improve clarity, focus, and accessibility. The revised version uses more recent and relevant references (2019–2024) and limits the number of sources to those directly supporting the research context. The problem statement, research question, and study rationale have been reorganized and simplified to clearly convey the importance and originality of the study. These changes are now visible in the tracked changes version of the manuscript.
Comment#4: Method
The material and methodology of the study are appropriate. However, there is a serious and important problem.
- Is the number of participants included in the study sufficient and appropriate for this study? How was the number calculated? Why should a power analysis be performed? I strongly recommend a power analysis.
- The non-tennis group seems very scattered, why was such a wide range preferred? It should be explained.
Authors Response#4_1: We appreciate the reviewer’s insightful comment and fully agree on the importance of ensuring an adequate sample size through statistical justification. At the time of data collection, the sample size (n = 39) was determined based on the availability of eligible athletes from local clubs and academies who met the inclusion criteria for both the tennis and non-tennis groups. However, we acknowledge that a formal a priori power analysis was not performed prior to recruitment. To address this, we have now conducted a post hoc power analysis using G*Power (version 3.1.9.7), based on the observed effect sizes for RMS activation differences between groups (e.g., d = 1.49 for lower trapezius at 90°). The achieved power (1–β) for this comparison exceeded 0.90, indicating a very high probability of detecting true differences with the current sample size.
Nevertheless, we recognize that conducting a priori power analysis is best practice to guide recruitment and avoid both Type II errors and overestimation of effect sizes. We have now added this limitation to the manuscript and clearly acknowledged it in the revised “Limitations” section (Section 4.1), along with the need for future studies to include power-based recruitment estimates. This clarification also reinforces the need for larger, stratified samples in future investigations to enhance generalizability.
Authors Response#4_2: We appreciate the reviewer’s observation. The heterogeneity of the non-tennis group was intentionally allowed to reflect a representative sample of athletes from non-overhead and non-unilateral sports disciplines. This diversity aimed to serve as a general comparison group without the specific neuromuscular adaptations commonly seen in tennis players. Participants were included only if they did not participate in racquet sports or any sport with predominant upper-limb dominance. While we acknowledge this introduces variability, it also enhances the ecological validity of the control group and avoids biasing the comparison toward any single non-tennis sport. This point has now been clarified in the Methods section and also addressed in the Limitations section of the manuscript.
Comment#5 Results:
- Abbreviations included in tables should be presented below the table. If ES represents effect size, statistical analyses may be incorrect. While p-value does not indicate a significant result, effect sizes are very large. All analyses should be reviewed.
- The findings were not well presented.
Authors Response#5_1:
We thank the reviewer for this important observation. We have carefully revised all tables to ensure that abbreviations—particularly “ES” (now explicitly labelled as "Effect Size (Cohen’s d)")—are clearly defined directly below each table. Additionally, we have reviewed the statistical analyses and confirm that the calculation and interpretation of effect sizes (Cohen’s d) were performed correctly. In cases where p-values did not reach the conventional threshold for significance (p > 0.05), we have emphasized the magnitude of the effect size in the Results and Discussion sections to provide a more comprehensive interpretation. This distinction has been clarified to avoid potential misinterpretation and to align with current recommendations for transparent statistical reporting. These adjustments are now reflected in the revised version of the manuscript.
Authors Response#5_2: Thank you, we have followed your recommendations and now it is amended.
Comment #6: Discussion
- In the first part of the discussion, it was not stated whether the hypothesis was accepted or not.
- The discussion section contains unnecessary details. It is written too long, too complicated. There is no integrity of the subject. Therefore, it is not understood. It should be rewritten and should include only current sources. It should also be short, simple and understandable for the purpose.
- In addition, every research has its strengths, weaknesses and limitations. These are not included in this research. In particular, there is no section on limitations. This is an important issue.
- In addition, no strong recommendations were made for some protic applications based on the research results.
Authors Response#6_1:
Hypothesis Clarification:
We have now explicitly addressed the study hypothesis at the beginning of the Discussion section. A clear statement has been added to confirm whether the hypothesis was supported by the data.
Authors Response#6_2:
Rewriting and Simplifying the Discussion:
The entire Discussion section has been substantially revised for clarity, focus, and conciseness. We removed redundant content and reorganized the narrative to improve coherence and alignment with the study objectives. Recent and relevant literature (2019–2024) has been prioritised, and only current sources are now cited.
Authors Response#6_3:
Limitations Section:
A detailed “Limitations” subsection has been added (Section 4.1), outlining the main methodological limitations of the study, including sample size, heterogeneity of the non-tennis group, and unmeasured variables such as training load. This section helps contextualize the findings and guide future research directions.
Authors Response#6_4:
Practical Applications:
We have strengthened the “Practical Applications” section by highlighting concrete recommendations for injury prevention programs in tennis. Specific exercise modalities, shoulder positions, and implementation strategies have been described for coaches and practitioners.
Reviewer 2 Report
Comments and Suggestions for Authors
Abstract
- explain what you mean by bilateral exercises for injury prevention
- The summary could benefit from greater quantitative precision.
- Check the number of keywords
Introduction
- It could be more synthetic in order not to lose the main focus of the research problem.
- Some concepts (such as muscle imbalances and GIRD) are explained several times in very similar ways.
Methods
- Review text formatting. Bold highlighting
- How did the participants access the study?
- the descriptive table should also provide data per group
- “randomized” is mentioned, but it is not clear how the assignment was made (no details on concealment or random sequencing).
- Inclusion and exclusion criteria by group
- How was the EMG signal filtered in the signal processing (also possible in the measuring device)? If so, what are the parameters of the filter?
- How long was the measurement duration of the EMG signal? How long did you wait to start the registration?
- The procedure for recording muscle activity should be more detailed.
Discussion and Conclusion
- The training load of the previous days was not considered, which may affect the EMG data.
- Several concepts are unnecessarily reiterated (e.g., explanation of 45° vs. 90° positions).
- Future lines of research: Although it mentions ideas, they could be better structured to guide future researchers.
- the conclusion should be more concrete and brief (some parts of the practical applications are repetitive with respect to the discussion).
Author Response
Comment #1: Abstract
- Explain what you mean by bilateral exercises for injury prevention
- The summary could benefit from greater quantitative precision
- Check the number of keywords
Authors Response #1:
- We have clarified in the abstract that bilateral exercises refer to shoulder prevention exercises performed with both arms simultaneously, targeting scapular stabilizers to promote symmetrical activation and reduce inter-limb imbalances.
- The abstract has been revised to include key quantitative findings (e.g., RMS values, peak activation differences, effect sizes) to increase specificity and highlight the strength of the observed outcomes.
- The number of keywords has been reduced and refined to six, ensuring they are specific and relevant to the study focus, in accordance with the journal’s guidelines.
Comment #2: Introduction
- It could be more synthetic in order not to lose the main focus of the research problem
- Some concepts (such as muscle imbalances and GIRD) are explained several times in very similar ways
Authors Response #2:
- Authors Response: The introduction has been restructured to eliminate tangential content, maintain focus on the central research problem, and provide a clear justification for the study.
- Authors Response: Repeated explanations of terms such as muscle imbalance and GIRD have been removed. The remaining definitions are concise and occur only once in the revised introduction to avoid redundancy.
Comment #3: Methods
- Review text formatting. Bold highlighting
Authors Response: Formatting inconsistencies have been corrected. Unnecessary bold text has been removed to improve readability and comply with journal style. - How did the participants access the study?
Authors Response: We have added a clarification stating that participants were recruited through direct contact with local tennis clubs and sports federations. Recruitment was voluntary and conducted through informed invitations. - The descriptive table should also provide data per group
Authors Response: Table 1 has been updated to include separate descriptive statistics for the tennis player and non-tennis player groups. - “Randomized” is mentioned, but it is not clear how the assignment was made (no details on concealment or random sequencing)
Authors Response: We have removed the word “randomized” from the description of the study design, as this was a cross-sectional comparison and not a randomized controlled trial. This correction clarifies the nature of the group allocation. - Inclusion and exclusion criteria by group
Authors Response: The inclusion/exclusion criteria have been reviewed and specified for each group in table 2. - How was the EMG signal filtered in the signal processing? What are the parameters of the filter?
Authors Response: Details regarding the EMG signal processing have been added. The system used an 8.4 Hz high-pass and 450 Hz low-pass bandpass filter, consistent with the mDurance® validation protocol [24]. - How long was the measurement duration of the EMG signal? How long did you wait to start the registration?
Authors Response: We have specified that each contraction was recorded for approximately 3–5 seconds, and data collection began after a 2-second stabilization phase to avoid noise at the start of the recording. - The procedure for recording muscle activity should be more detailed
Authors Response: We have expanded the description of the EMG setup, including preparation steps (skin cleaning, electrode placement following SENIAM guidelines), posture during tasks, and breath coordination to standardize recording conditions.
Comment #4: Discussion and Conclusion
- The training load of the previous days was not considered, which may affect the EMG data
Authors Response: This limitation has been acknowledged and discussed in Section 4.4. The potential influence of prior training load and neuromuscular fatigue on EMG results is now clearly addressed. - Several concepts are unnecessarily reiterated (e.g., explanation of 45° vs. 90° positions)
Authors Response: Repetitive discussion of shoulder abduction angles has been removed. Only the essential biomechanical rationale is maintained to improve clarity and reduce redundancy. - Future lines of research: Although it mentions ideas, they could be better structured to guide future researchers
Authors Response: The “Future Directions” portion of the discussion has been reformulated into a clearer structure with bullet points suggesting: (a) longitudinal tracking, (b) testing other scapular muscles, and (c) stratifying by age/training level. - The conclusion should be more concrete and brief (some parts of the practical applications are repetitive with respect to the discussion)
Authors Response: The conclusion has been rewritten to be more concise. Redundant practical applications already discussed in the main text were removed, and key take-home messages now reflect the core findings of the study more succinctly.
Round 2
Reviewer 1 Report
Comments and Suggestions for Authors
Many of the edits expressed have been made. Thank you. No further comments.
Reviewer 2 Report
Comments and Suggestions for Authors
Most of the reviewer's input and suggestions have been addressed.